# Structural Augmentation in Seismic Data for Fault Prediction

Shenghou Wang [1], Xu Si [2,*], Zhongxian Cai [1] and Yatong Cui [3]

1   Key Laboratory of Tectonics and Petroleum Resources of Ministry of Education, China University of Geosciences (Wuhan), Wuhan 430074, China
2   School of Earth and Space Sciences, University of Science and Technology of China, Hefei 230026, China
3   Tianjin Survey Design Institute Group Co., Ltd., Tianjin 300191, China
*   Correspondence: xusi@mail.ustc.edu.cn

**Abstract:** Fault interpretation tasks become more and more difficult as the complexity of seismic exploration increases, especially for ultra-deep seismic data. Recently, numerous researchers have utilized automatic interpretation techniques based on deep learning to improve the efficiency and accuracy of fault prediction. Although deep learning methods have powerful data information processing capabilities, the applicability of deep neural networks may still be limited by the range of learned information. Therefore, we develop a new technique called structural data augmentation to enhance the diversity of the datasets. Concretely, we utilize different geological structure theories to incorporate virtual folds and faults in the field seismic data to improve the diversity and generalization ability of the training datasets. To cope with the multi-stage and multi-scale complex structures developed in ultra-deep strata, the proposed augmentation workflow increases data diversity by generating various virtual structures containing multi-scale folds, listric faults, oblique-slip displacement fields, and multi-directional fault drags. Tests on the field seismic data show that our method not only outperforms conventional seismic attributes but also has advantages over other machine learning methods.

**Keywords:** fault detection; deep learning; training sets; data augmentation; geological structure

## 1. Introduction

As a typical component of the geological structure, faults can serve as potential channels for oil and gas migration and accumulation [1–3] or reveal the spatial distribution and heterogeneity of subsurface targets [4,5]. Therefore, fault identification methods have become indispensable technologies in seismic data interpretation [6,7]. However, the complex structures and low-quality information contained in seismic data become unavoidable challenges, especially for ultra-deep fault interpretation [8]. To accurately and effectively identify faults from seismic data, many researchers have proposed various practical strategies.

In the early days, researchers used the human–computer interaction method to manually interpret the seismic reflection discontinuity caused by the fault [9–11]. However, using only seismic signals to manually identify faults often produces strong human-oriented recognition results, and the interpretation results are severely limited by the signal-to-noise ratio and seismic data resolution. Therefore, the joint well-seismic interpretation technologies are derived [9,12,13]. Through the combination of log and seismic data, the accuracy of fault interpretation is effectively improved. However, these fault identification technologies rely on manual fault labeling, which is highly time-consuming, especially for 3D seismic data.

Therefore, some semi-automatic methods [14,15] and automatic fault recognition methods [16–18] have been proposed to reduce the workload of manual interpretation. These methods suggest that the purpose of automatic fault identification can be achieved by calculating and highlighting the discontinuity of seismic events. Coherence attribute as a classic

automatic fault identification technology is widely used in seismic data interpretation, and many coherence-based algorithms have been developed, such as the first-generation coherence algorithm [16], the least-square coherence algorithm [17], the semblance-based coherence algorithm [18], the eigenstructure-based coherence algorithm [19], the frequency-decomposed coherence algorithm [20], the predictive coherence algorithm [21], and the structure-oriented coherence [22]. In addition to coherent attributes, curvature attributes are also commonly used in seismic data interpretation. Since Richard [23] proposed the Gaussian Curvature Analysis (GCA) method to detect abnormal strain regions in the structures, many curvature-based algorithms have been discussed, such as surface curvature algorithm [24], volumetric curvature algorithm [25], curvature gradient [26], and flexure attribute [27,28].

However, the fault identification methods based on the discontinuity of the reflection events are sensitive to seismic noise [29,30]. Therefore, many researchers have developed fault identification methods based on anti-noise or fault structure enhancement [31–36], such as ant tracking [37,38], spectral coherence [20], multi-azimuth coherence [39], fault voting [40], and multi-sensitive attribute fusion [41–43]. However, these enhanced recognition methods with structural constraints may still produce erroneous recognition results, especially in the data area where faults are adjacent to each other [44]. To further improve the accuracy of fault prediction, artificial intelligence algorithms are introduced in this paper to identify faults.

Artificial intelligence methods have surprising interdisciplinary applicability [45–50] and have been introduced into the field of geosciences [51,52], such as seismic imaging [53], seismic waveform classification [54], event localization [55,56], earthquake prediction [57,58], and earthquake warning [59].

Recently, some researchers began to use artificial intelligence to identify faults and obtained effective results. These methods can be mainly divided into three categories: The first category of methods trains the neural networks with field seismic data and manually interpreted labels [60–63]. However, the confidentiality and scarcity of field and label data often hinder the application of machine learning methods [51,52,64]. Therefore, the second type of methods use synthetic seismic data and theoretical fault labels to train the neural network [44,65–73]. These methods effectively alleviate the problem of training datasets scarcity and excel in some seismic data applications. However, the geological information and waveform features contained in synthetic data are usually ideal and bounded. It is difficult to fully bridge the gap between synthetic and field data using only theoretical models [62,74]. The third category of methods utilizes the idea of transfer learning to identify faults [74–78]. A small amount of field seismic data and manually interpreted labels are used to fine-tune the neural network pre-trained from synthetic data. While this fine-tuning technique can improve the performance of fault detection, the results are not the best in some field data applications [62].

In summary, the current popular machine-learning-based fault detection methods may all face the problem of insufficient data and information. One of the biggest challenges for machine learning in fault identification and even in geosciences is the lack of numerous, informative, and field-like training data [51,52]. Some classic data augmentation methods can help alleviate this difficulty [62,79,80], such as flipping [81], rotating [82], cropping [83], intensity transformation [84], down-sampling [85], random brightness [86], grid distortion [62], and random drop [87]. However, some of these data augmentation methods may not meet the geological constraints when applied directly to seismic data. Therefore, we attempt to bridge the gap between the training set and the field seismic data to improve the fault characterization capability in this paper.

In this paper, the characterization of fault structures in 3D seismic data is considered as an image segmentation problem, which is solved by a simple 3D U-net. The architecture of the 3D U-net is improved from the 2D U-net used in the medical image field [88]. Although there are many networks with more parameters and more complex architectures now [89–93], we still only apply a simple U-shaped network in this study. The simple 3D

Unet network architecture is used because 3D U-net has been validated in fault detection tasks [44,66,70] and because we want to put more emphasis on the impact of the data on the results and improve computational efficiency.

To alleviate the problem of insufficient labeled field-like datasets in machine learning methods, we propose a structural data augmentation method in this study. This method creates virtual geometric folds and faults in field seismic signals and interpretation data to obtain semi-real-semi-synthetic seismic data and corresponding labels. Compared to methods that train networks using field data and manually interpreted data, our approach allows training datasets to contain not only the experience of geologists but also the knowledge of theoretical models. In addition to improving the generalization capability, our method requires interpretation for only a small amount of field data, which avoids the tedious work of completely labeling large seismic data. Compared to synthetic data learning and transfer learning methods, our method is applied directly to field data, allowing our approach to be more effective in bridging the difference between the training set and field data.

Through our structural data augmentation framework, we automatically generate 200 data–label pairs. After further rotation augmentation, these data sets effectively train our neural network to identify fault structures in 3D seismic signals. To evaluate the effectiveness of our method, we compare two classical attribute methods (coherence and curvature) and three machine learning methods (manual interpretation data learning, synthetic data learning, and transfer learning) with the proposed approach through several field data examples. The fault detection results from field seismic data demonstrate that our method not only outperforms conventional seismic attributes but also has advantages over the other three types of machine learning methods.

## 2. Methods

When training a deep learning network to detect fault structures in 3D seismic data, numerous field-like 3D seismic volumes are required as inputs, and the corresponding 3D fault structures are required as labels. It is not difficult to obtain numerous seismic volumes by augmenting field data with existing data augmentation techniques derived from computer vision [62,79,80]. However, to effectively improve the training performance, the applied data augmentation techniques cannot change the semantics of the data [94]. Conventional data augmentation methods, such as flipping [81], random brightness [86], random drop [87], and grid distortion [62], may lead to stratigraphic overturn and tectonic distortion, which can produce geologically unacceptable data samples. Therefore, some of these data augmentation algorithms may only provide limited assistance for fault detection.

To cope with the multi-stage and multi-scale complex structures developed in ultra-deep seismic data, the proposed augmentation workflow increases data diversity by generating various virtual structures containing multi-scale folds, listric faults, oblique-slip displacement fields, and multi-directional fault drags. The basic idea of this augmentation method is to add virtual faults and folds into the seismic data to ensure that the data remain reasonable in the geological sense. For structural geology, this process can be considered as the development of folds or faults due to multi-stage tectonic events. Randomly adding various folds and faults into the 3D field seismic data also increases the diversity and generalization ability of the samples. Figure 1 demonstrates an example of adding a series of virtual structures into the field seismic data and manual fault interpretation. The detailed workflow of the structural data augmentation is as follows:

(1) As shown in Figure 1a,d, sub-volumes ($128 \times 128 \times 128$ in this study) are truncated from a 3D field seismic data volume and corresponding manually interpreted volume.

(2) Generate virtual folds. As shown in Figure 1b, the process of adding virtual folds can be simulated by raising and lowering the data points. The randomly generated folds determine the shift distance of each data point.

(3) Generate virtual faults. As shown in Figure 1c, the process of adding virtual faults can be simulated by making discontinuities on both sides of virtual fault surfaces.

The randomly generated fault surfaces determine the specific location and basic shape of faults. The randomly simulated fault near-field displacements and fault drags determine the direction and magnitude of the discontinuities on both sides of the faults. To further increase the generalization ability of the dataset and the anti-noise ability of fault identification, a small amount of random noise is also added into the seismic data ($SNR = [4,5]$ in this study).

(4) Repeat steps 2 and 3 until the generated samples are sufficient to train a deep learning network model.

Different from conventional data generation workflows [44,66–68], the proposed structural data augmentation workflow can automatically generate various datasets containing multi-scale folds, listric faults, and oblique-slip displacement fields to provide a variety of data-label pairs to train neural networks.

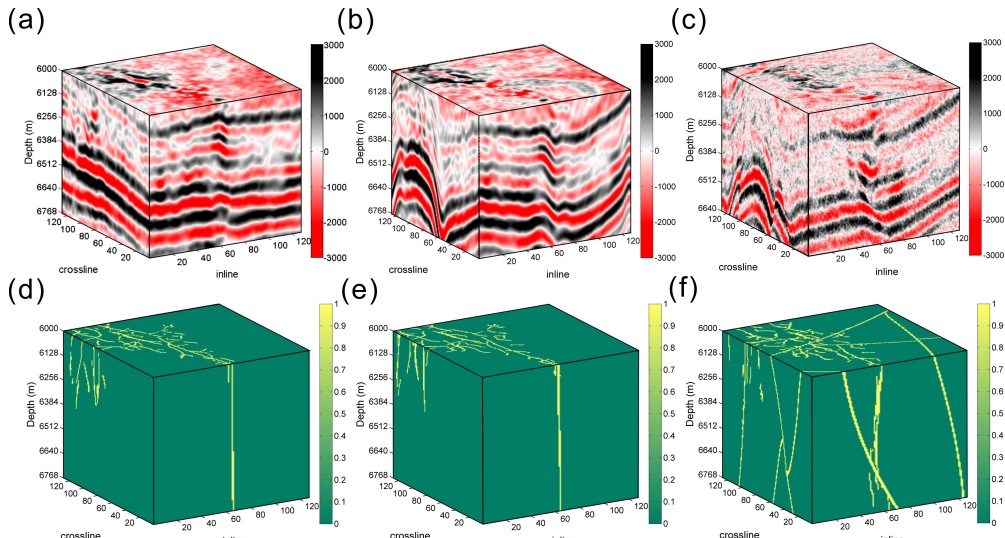

**Figure 1.** The workflow of structural data augmentation. (**a**) Raw 3D seismic data. (**b**) Seismic data after adding virtual folds. (**c**) Seismic data after adding virtual faults and random noise. (**d**) Manual fault interpretation label data. (**e**) Label data after adding virtual folds. (**f**) Label data after adding virtual faults.

## 2.1. Adding Virtual Folds

The simulation of the virtual fold structures is divided into three parts in this study: large-scale folds $S_1(X, Y, Z)$, local folds $S_2(X, Y, Z)$, and dips $S_3(X, Y, Z)$.

Unlike conventional methods using standard Fourier series to model folds [95], elliptic equations are applied in Fourier series to increase the asymmetry in this study. Concretely, the standard Fourier series are diverse in the $Z$-direction, but they maintain symmetry in the $X$- or $Y$-direction (Figure 2a). As shown in Figure 2b, the generated large-scale folds can be asymmetric in the lateral direction through the following equation:

$$S_1(X,Y,Z) = A_0 + \left(1 - \left|\frac{a(X-X_0)+b(Y-Y_0)}{X_{max}+Y_{max}}\right|\right) \sum_{k=1}^{N}\left(A_k \cos\left(\frac{k\pi E(X,Y)}{\lambda_k}\right) + B_k \sin\left(\frac{k\pi E(X,Y)}{\lambda_k}\right)\right)$$

$$E(X,Y) = \sqrt{\left(\frac{\cos\varphi_k(X-X_0) - \sin\varphi_k(Y-Y_0)}{L[x]_k}\right)^2 + \left(\frac{\sin\varphi_k(X-X_0) + \cos\varphi_k(Y-Y_0)}{L[y]_k}\right)^2} \tag{1}$$

where $A_k$ and $B_k$, and $\lambda_k$ are the basis parameters in the standard Fourier series, controlling the amplitude and period of the trigonometric functions, respectively. The elliptic long axis $L[x]_k$, elliptic short axis $L[y]_k$, and rotation angle $\varphi_k$ control the variation of the virtual folds in the lateral direction. By applying the linear decay function $1 - \left|\frac{a(X-X_0)+b(Y-Y_0)}{X_{max}+Y_{max}}\right|$, the amplitude of the large-scale folds gradually decreases from the center to the edges. In this study, we assume that large virtual folds belong to a single stratigraphic unit and

have nearly uniform surfaces. If necessary, additional parameters can be applied in the Z-direction to increase diversity.

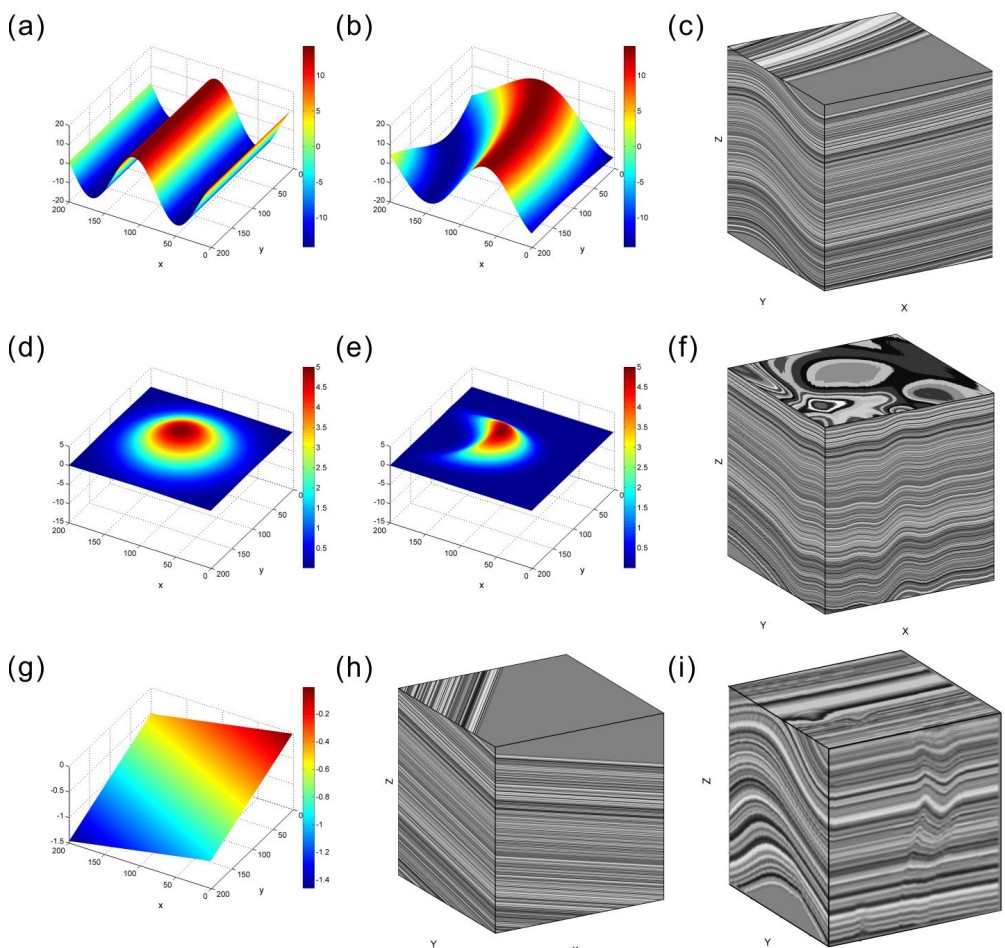

**Figure 2.** Demonstration of virtual folds. (**a**) A large-scale fold generated by the standard Fourier series. (**b**) A large-scale fold generated in this study. (**c**) A structure model with large-scale folds. (**d**) A local fold generated by the standard Gaussian function. (**e**) A local fold generated in this study. (**f**) A structure model with local folds. (**g**) Dips generated in this study. (**h**) A structure model with dips. (**i**) A structure model with multi-scale folds.

Different from the symmetric standard Gaussian equation used by Wu et al. [68], we use the following equation to generate asymmetric local virtual folds (Figure 2d):

$$S_2(X,Y,Z) = \begin{array}{l} c\frac{Z}{Z_{max}}\sum_{k=1}^{N}D_k\exp\left[-\left(\frac{\cos\phi_k(X-E_k)-\sin\phi_k(Y-F_k)}{2\sigma[x]_k}\right)^2 \\ -\left(\frac{\sin\phi_k(X-E_k)+\cos\phi_k(Y-F_k)}{2\sigma[y]_k}\right)^2\right], \end{array} \tag{2}$$

where the parameters $(E_k, F_k)$ and $D_k$ control the center and amplitude of the local folds, respectively. As shown in Figure 2e, by controlling the parameters of long axis $\sigma[x]_k$, short axis $\sigma[y]_k$, and rotation angle $\phi_k$, the generated local folds are not limited to symmetric circles.

As shown in Figure 2g, the linear function that generates the dip structure uses parameters $G$ and $H$ to control the dip of the formation in the X- and Y-directions:

$$S_3(X,Y,Z) = GX + HY + l. \tag{3}$$

As shown in Figure 1b, the folded data $D_1(X, Y, Z)$ can be calculated from the field seismic data $D_0(X, Y, Z)$ by the defined virtual fold fields $S_1(X, Y, Z)$, $S_2(X, Y, Z)$ and $S_3(X, Y, Z)$:

$$D_1(X, Y, Z) = D_0(X, Y, Z + S_1(X, Y, Z) + S_2(X, Y, Z) + S_3(X, Y, Z)). \tag{4}$$

The pattern of virtual folds is defined by different combinations of many parameters. These randomly selected parameters can generate a variety of unique virtual folds for structural data augmentation.

### 2.2. Adding Virtual Faults

As shown in Figure 3, the simulation of the virtual fault structures is divided into four parts in this study: fault reference plane, fault near-field displacement, fault surface, and fault drag. The initial reference plane of a fault can be defined by the reference point $(X_0, Y_0, Z_0)$, azimuth $\Phi$, and dip angle $\Theta$ [96]:

$$\begin{bmatrix} \bar{X} \\ \bar{Y} \\ \bar{Z} \end{bmatrix} = \mathbb{R} \begin{bmatrix} X - X_0 \\ Y - Y_0 \\ Z - Z_0 \end{bmatrix} \quad \mathbb{R} = \begin{bmatrix} \sin\Phi & \cos\Phi & 0 \\ \cos\Phi\cos\Theta & -\sin\Phi\cos\Theta & \sin\Theta \\ \cos\Phi\sin\Theta & \sin\Phi\sin\Theta & -\cos\Theta \end{bmatrix}, \tag{5}$$

where $\mathbb{R}$ is the coordinate rotation matrix, which defines the normal direction of the reference plane as $\bar{Z}$, the strike direction as $\bar{X}$, and the dip direction as $\bar{Y}$ (Figure 3a).

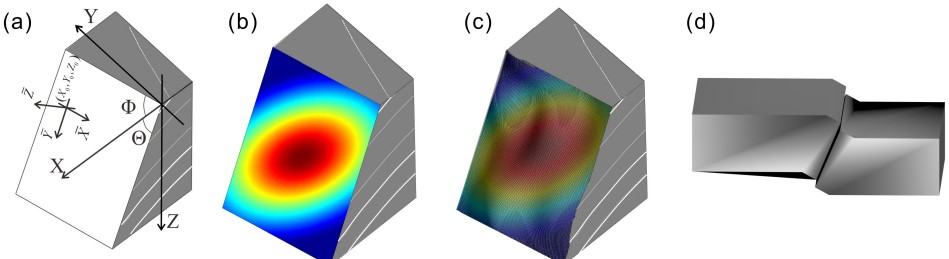

(a)　　　　(b)　　　　(c)　　　　(d)

**Figure 3.** Four parts of virtual fault structures. (**a**) Reference plane. (**b**) Near-field displacement. (**c**) Fault surface. (**d**) Fault drag.

### 2.2.1. Fault Displacement

Different from conventional methods [68,96], two additional parameters are applied to the fault displacement simulation in this study (Figure 4): the rotation angle of the fault displacement $\Psi_{dis}$ and the rotation angle of the fault attenuation ellipse $\Psi$. In nature, there are not only many dip-slip faults [97] but also many strike-slip faults [98]. Therefore, different strike-slip and dip-slip displacements can be combined on a fault surface to produce arbitrary oblique-slip directions. In addition, although the attenuation of fault displacement is widely considered to satisfy an elliptic equation [99–101], it has also been shown that the attenuation ellipse may have different rotation angles in different faults [102,103]. We therefore model the near-field displacement of faults by the following equation:

$$d(\bar{X}, \bar{Y}) = 2Amp_d(1 - r(\bar{X}, \bar{Y}))\sqrt{\left(\frac{1 + r(\bar{X}, \bar{Y})}{2}\right)^2 - r(\bar{X}, \bar{Y})^2}$$

$$r(\bar{X}, \bar{Y}) = \sqrt{\left(\frac{\cos\Psi(\bar{X} - \bar{X}_0) - \sin\Psi(\bar{Y} - \bar{Y}_0)}{L[\bar{x}]}\right)^2 + \left(\frac{\sin\Psi(\bar{X} - \bar{X}_0) + \cos\Psi(\bar{Y} - \bar{Y}_0)}{L[\bar{y}]}\right)^2} , \tag{6}$$

$$d(\bar{X}, \bar{Y})[\bar{x}], = d(\bar{X}, \bar{Y})\sin\Psi_{dis} \qquad d(\bar{X}, \bar{Y})[\bar{y}] = d(\bar{X}, \bar{Y})\cos\Psi_{dis}$$

where the parameters $(\bar{X}_0, \bar{Y}_0)$, $(L[\bar{x}], L[\bar{y}])$ and $Amp_d$ are the classical parameters of the displacement attenuation equation, controlling the center, radius, and amplitude of the

near-field displacement, respectively. The parameters $d(\bar{X}, \bar{Y})[\bar{x}]$ and $d(\bar{X}, \bar{Y})[\bar{y}]$ represent the components of displacement in the strike-slip direction $\bar{X}$ and the dip-slip direction $\bar{Y}$, respectively. On the basis of these classical parameters, numerous displacement patterns of oblique-slip faults can be simulated by different combinations of displacement rotation angles $\Psi_{dis}$ and attenuation ellipse rotation angles $\Psi$.

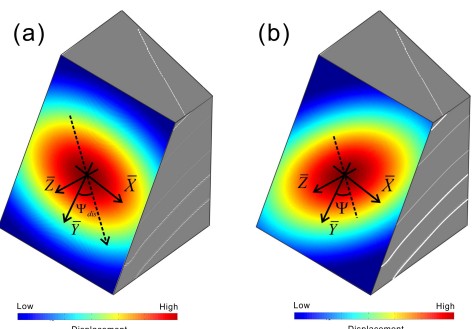

**Figure 4.** Demonstration of fault displacement distribution. (**a**) Near-field displacement with displacement rotation angle $\Psi_{dis}$. (**b**) Near-field displacement with attenuation rotation angle $\Psi$. The dashed arrow indicates the direction of displacement, and the dotted line indicates the direction of the short axis of the attenuation ellipse.

### 2.2.2. Fault Surface

As shown in Figure 5, the generation of fault surfaces is mainly divided into two parts in this study: ellipsoidal surface $f_1(\bar{X}, \bar{Y})$ and random perturbation $f_2(\bar{X}, \bar{Y})$.

$$f(\bar{X}, \bar{Y}) = f_1(\bar{X}, \bar{Y}) + f_2(\bar{X}, \bar{Y}). \tag{7}$$

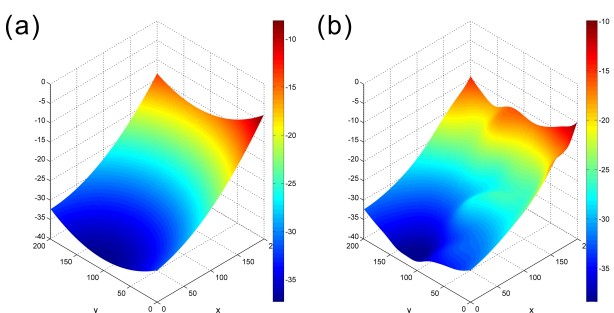

**Figure 5.** Generation of fault surfaces. (**a**) Ellipsoidal surface. (**b**) Random perturbation.

Unlike the conventional methods that use only the random simulation algorithms to generate faults [68,104,105], which has a low probability of generating listric faults, we apply an ellipsoidal surface equation to create listric faults with different shapes:

$$f_1(\bar{X}, \bar{Y}) = \left[ 1 - \left( \frac{X_R}{L[x_R]} \right)^2 - \left( \frac{Y_R}{L[y_R]} \right)^2 - \left( \frac{Z_R}{L[z_R]} \right)^2 \right] Amp_E$$

$$\begin{bmatrix} X_R \\ Y_R \\ Z_R \end{bmatrix} = R_{\bar{x}} R_{\bar{y}} R_{\bar{z}} \begin{bmatrix} \bar{X} - \bar{X}_0 \\ \bar{Y} - \bar{Y}_0 \\ \bar{Z}_0 \end{bmatrix} \tag{8}$$

$$R_{\bar{x}}(\theta_{\bar{x}}) = \begin{bmatrix} 1 & 0 & 0 \\ 0 & \cos\theta_{\bar{x}} & -\sin\theta_{\bar{x}} \\ 0 & \sin\theta_{\bar{x}} & \cos\theta_{\bar{x}} \end{bmatrix}, R_{\bar{y}}(\theta_{\bar{y}}) = \begin{bmatrix} \cos\theta_{\bar{y}} & 0 & \sin\theta_{\bar{y}} \\ 0 & 1 & 0 \\ -\sin\theta_{\bar{y}} & 0 & \cos\theta_{\bar{y}} \end{bmatrix}, R_{\bar{z}}(\theta_{\bar{z}}) = \begin{bmatrix} \cos\theta_{\bar{z}} & -\sin\theta_{\bar{z}} & 0 \\ \sin\theta_{\bar{z}} & \cos\theta_{\bar{z}} & 0 \\ 0 & 0 & 1 \end{bmatrix}$$

where $(\bar{X}_0, \bar{Y}_0, \bar{Z}_0)$, $(L[x_R], L[y_R], L[z_R])$, and $(\theta_{\bar{x}}, \theta_{\bar{y}}, \theta_{\bar{z}})$ are the classical parameters of the standard ellipsoidal surface equation, controlling the center, radius, and rotation angle of the ellipsoidal surface, respectively. In some fault examples (Figure 6a,b), there are local

structures with uniform orientation, such as ridge-and-groove morphology [106–108], so we control the random perturbation of the fault surface by the following equation:

$$f_2(\bar{X}, \bar{Y}) = \sum_{k=1}^{N} Amp_k \exp\left[ -\left( \frac{\cos\Psi_k(\bar{X} - J_k) - \sin\Psi_k(\bar{Y} - K_k)}{2\sigma[\bar{x}]_k} \right)^2 - \left( \frac{\sin\Psi_k(\bar{X} - J_k) + \cos\Psi_k(\bar{Y} - K_k)}{2\sigma[\bar{y}]_k} \right)^2 \right], \tag{9}$$

where the parameters $(J_k, K_k)$, $(\sigma[\bar{x}]_k, \sigma[\bar{y}]_k)$ and $Amp_k$ control the center, radius, and magnitude of the local perturbation, respectively. The random perturbation of the fault can be made consistent with the fault displacement direction by controlling the rotation angle $\Psi_k$ (Figure 6c).

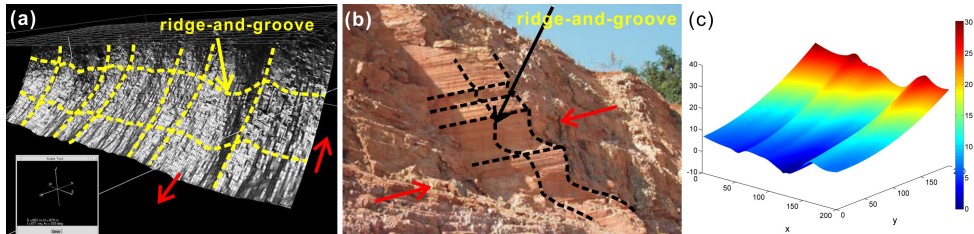

**Figure 6.** Demonstrations of ridge-and-groove morphology. (**a**) Normal fault surface modified after [108]. (**b**) Strike-slip fault surface modified after [109]. (**c**) Ridge-and-groove morphology generated in this paper. The red arrows represent the fault displacement direction. Reprinted with permission from [109]. 2022, John Wiley and Sons.

With the calculation of $f_1(\bar{X}, \bar{Y})$ and $f_2(\bar{X}, \bar{Y})$, numerous curved fault surfaces can be generated, and the insufficiency of random seed methods in listric fault simulation can be avoided.

### 2.2.3. Fault Drag

As shown in Figure 7, the fault drag is described in two types in this study: reverse drag and normal drag. For the reverse drag simulation, the following nonlinear equation is popularly used [110]:

$$\alpha(\bar{X}, \bar{Y}, \bar{Z}) = \left( 1 - \frac{|\bar{Z} - f(\bar{X}, \bar{Y})|^2}{R} \right) \qquad f(\bar{X}, \bar{Y}) - R \le \bar{Z} \le f(\bar{X}, \bar{Y}) + R, \tag{10}$$

where $f(\bar{X}, \bar{Y})$ is the fault surface defined in Equation (7), and $R$ is the drag radius.

Inspired by the sigmoid function [111], the normal drag is simulated using the following truncated sigmoid function:

$$\alpha(\bar{X}, \bar{Y}, \bar{Z}) = \begin{cases} \frac{2}{1+\exp\left( \frac{-\mu(\bar{Z}-f(\bar{X},\bar{Y})-\delta)}{R} \right)} - 1 & f(\bar{X}, \bar{Y}) - R \le \bar{Z} \le f(\bar{X}, \bar{Y}) \\ \frac{2}{1+\exp\left( \frac{-\mu(\bar{Z}-f(\bar{X},\bar{Y})+\delta)}{R} \right)} - 1 & f(\bar{X}, \bar{Y}) \le \bar{Z} \le f(\bar{X}, \bar{Y}) + R, \\ 1 & |\bar{Z} - f(\bar{X}, \bar{Y})| \ge R \end{cases} \tag{11}$$

where the truncation distance $\delta$ controls the displacement at the drag center, and the scaling factor $\mu$ controls the shape of the drag.

Different from the conventional methods [68,96], we apply the drag rotation angle $\Psi_{drag}$ to control the generation of fault drags. In nature, fault drag exists not only with dip-slip faults [112–114] but also with strike-slip faults [115–117]. Therefore, different strike-slip and dip-slip drags can be combined to form arbitrary drag directions on a fault surface (Figure 7):

$$\begin{cases} D[\bar{x}](\bar{X}, \bar{Y}, \bar{Z}) = \lambda\alpha(\bar{X}, \bar{Y}, \bar{Z})d(\bar{X}, \bar{Y})[\bar{x}]\sin\Psi_{drag} & f(\bar{X}, \bar{Y}) \leq \bar{Z} \leq f(\bar{X}, \bar{Y}) + R \\ D[\bar{x}](\bar{X}, \bar{Y}, \bar{Z}) = (\lambda - 1)\alpha(\bar{X}, \bar{Y}, \bar{Z})d(\bar{X}, \bar{Y})[\bar{x}]\sin\Psi_{drag} & f(\bar{X}, \bar{Y}) - R \leq \bar{Z} \leq f(\bar{X}, \bar{Y}) \\ D[\bar{y}](\bar{X}, \bar{Y}, \bar{Z}) = \lambda\alpha(\bar{X}, \bar{Y}, \bar{Z})d(\bar{X}, \bar{Y})[\bar{y}]\cos\Psi_{drag} & f(\bar{X}, \bar{Y}) \leq \bar{Z} \leq f(\bar{X}, \bar{Y}) + R \\ D[\bar{y}](\bar{X}, \bar{Y}, \bar{Z}) = (\lambda - 1)\alpha(\bar{X}, \bar{Y}, \bar{Z})d(\bar{X}, \bar{Y})[\bar{y}]\cos\Psi_{drag} & f(\bar{X}, \bar{Y}) - R \leq \bar{Z} \leq f(\bar{X}, \bar{Y}) \end{cases}, \tag{12}$$

where the components of displacement $d(\bar{X}, \bar{Y})[\bar{x}]$ and $d(\bar{X}, \bar{Y})[\bar{y}]$ are mentioned in Equation (6), $R$ is the drag radius, and $\lambda(0 \leq \lambda \leq 1)$ represents the displacement ratio of hanging wall and footwall.

(a)    (b)

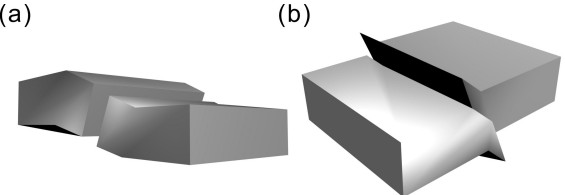

**Figure 7.** Demonstration of oblique-slip fault drag. (**a**) Normal drag in sinistral-reverse fault. (**b**) Reverse drag in sinistral-reverse fault.

To ensure the rationality of the hanging wall and foot wall on both sides of the fault surface, the displacement in the normal direction of the fault surface is defined by the following formula:

$$D(\bar{z})(\bar{X}, \bar{Y}, \bar{Z}) = f(\bar{X} + D[\bar{x}](\bar{X}, \bar{Y}, \bar{Z}), \bar{Y} + D[\bar{y}](\bar{X}, \bar{Y}, \bar{Z})) - f(\bar{X}, \bar{Y}). \tag{13}$$

Finally, after calculating the three components of the displacement field, we can generate virtual faults by the following equation:

$$\begin{bmatrix} \tilde{X} \\ \tilde{Y} \\ \tilde{Z} \end{bmatrix} = \mathbb{R}^{-1}\left( \begin{bmatrix} \bar{X} \\ \bar{Y} \\ \bar{Z} \end{bmatrix} + \begin{bmatrix} D[\bar{x}](\bar{X}, \bar{Y}, \bar{Z}) \\ D[\bar{y}](\bar{X}, \bar{Y}, \bar{Z}) \\ D[\bar{z}](\bar{X}, \bar{Y}, \bar{Z}) \end{bmatrix} \right) + \begin{bmatrix} X_0 \\ Y_0 \\ Z_0 \end{bmatrix}, \tag{14}$$

where the rotation factor $\mathbb{R}$ is mentioned in Equation (5). By arbitrarily combining the parameters of near-field displacement $d(\bar{X}, \bar{Y})$, fault surface $f(\bar{X}, \bar{Y})$, and fault drag $\alpha(\bar{X}, \bar{Y}, \bar{Z})$, many unique virtual faults can be generated. As shown in Figure 1c, numerous virtual faults including linear faults, listric faults, oblique-slip faults, and normal and reverse drag faults can be generated through a series of calculations. By adding a variety of virtual faults and folds, a highly diverse field-like dataset can be obtained.

## 3. Seismic Fault Detection Based on U-Net

In this work, we apply a simple 3D U-net to identify faults from 3D seismic volumes. The simple 3D U-net network architecture is used because we want a fair comparison with the work of Wu et al. [68] and because we want to put more emphasis on the impact of the data on the results. In the process of training and updating the neural network, the Adam method and the mean squared error (MSE) are used to optimize the network parameters.

### 3.1. Neural Network Architecture

Our network architecture is modified from a 2D U-shaped network used to implement 2D medical image segmentation [88], in which the input seismic data are first downsampled and feature extracted by an encoder, and then, they are up-sampled by a decoder to map the features into fault zones. As shown in Figure 8, our network utilizes skip connections to efficiently aggregate multi-scale semantic features from shallow and deep layers in the encoder. These skip connections can effectively reduce the long-path dependence of the network and transfer feature details between the encoder and decoder, which helps to improve the accuracy and computational efficiency of network learning.

In our network architecture, the encoder part consists of seven 3D convolutional layers in total. After inputting 3D seismic data, we apply successive convolutional blocks for feature extraction and down-sampling, which compresses the computational cost while increasing the receptive field. Each convolutional block contains two 3 × 3 × 3 convolutional layers, two rectified linear unit (ReLU) layers, and one 3D maxpooling layer. As shown in Figure 8, multi-scale semantic information is gradually extracted from shallow and deep networks through three convolutional blocks, and each scale is composed of two 3D convolutional layers.

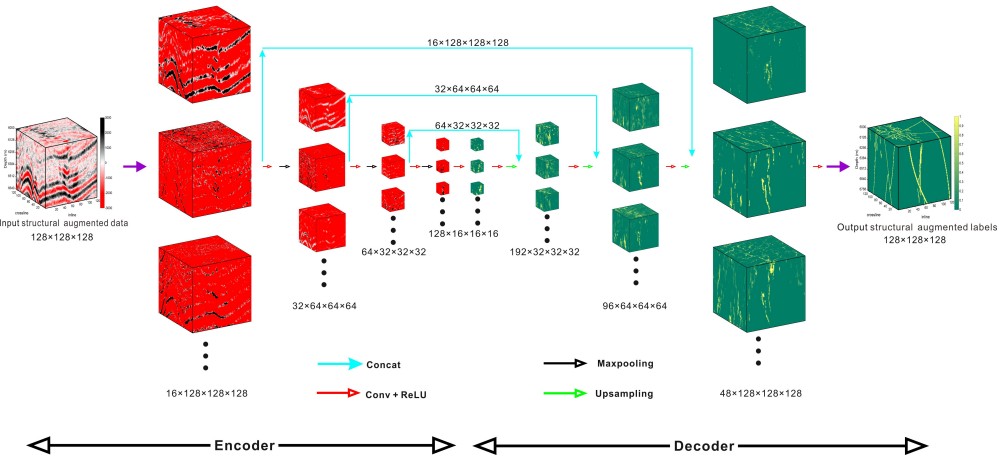

**Figure 8.** A deep convolutional neural network for fault detection.

The decoder, consisting of three convolutional blocks and three transposed convolutional layers, upsamples and maps features into fault labels with the same size as the 3D seismic volume. The network layers at each scale are also composed of two 3 × 3 × 3 convolutional layers and two ReLu layers.

### 3.2. Loss Function

The estimation of fault zones can be considered as an image segmentation problem, and the MSE is a common evaluation metric used to measure the difference between predictions and labels as follows:

$$Loss = \frac{1}{N} \sum_{k=1}^{N} (y_k - p_k)^2,$$ (15)

where $N$ represents the number of pixels in the input seismic data. The predicted fault probability $p_k$ and theoretical fault label $y_k$ at each pixel measure the computational error of the updated neural network.

### 3.3. Network Training

As shown in Figure 9, to further compare and validate the effectiveness of the structural data augmentation framework, our workflow is compared with three recently popular machine learning approaches (manually interpreted data learning, synthetic data learning, and transfer learning).

#### 3.3.1. Network Training Based on Manual Interpretation Data

In recent years, training neural networks for fault recognition using manually interpreted data has been considered as an effective way [60–63]. However, the confidentiality and scarcity of field data and interpretation results often hinder the application of machine learning methods [51,52]. Therefore, some classical augmentation methods in the field of computer vision are often utilized to augment field data [62,79,80]. In this paper, we truncate a 400 × 400 × 400 sub-volume and the corresponding manually interpreted

data from the work area A (Figure 10a). Similar to An et al. [62,79], 800 data–label pairs (128 × 128 × 128 in this study) are obtained using the classical augmentation method to train the neural network Net_manual (Figure 9a).

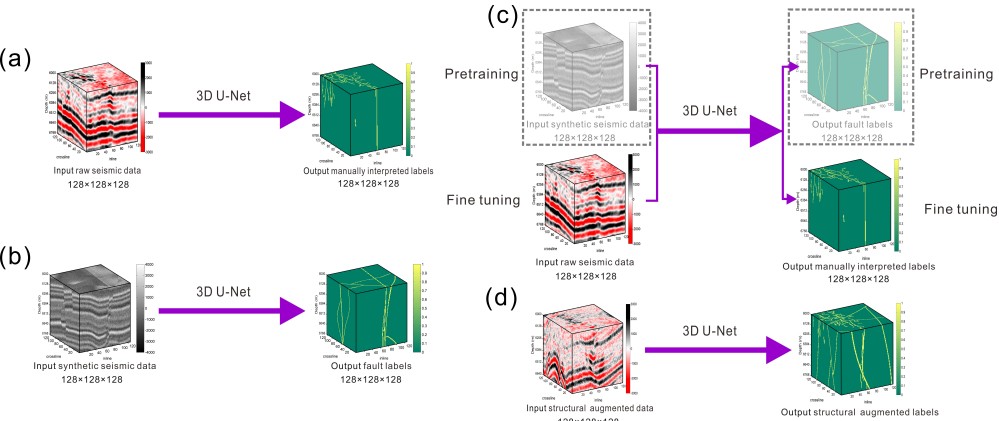

**Figure 9.** Network training based on (**a**) manual interpretation data, (**b**) synthetic data, (**c**) transfer learning, and (**d**) structural data augmentation.

### 3.3.2. Network Training Based on Synthetic Data

The second type of methods use synthetic seismic data and theoretical fault labels to train the neural network [68,70,73]. These methods effectively alleviate the problem of training datasets scarcity and excel in some seismic data applications. In this paper, we make some minor changes based on the work of Wu et al. [68] because these changes do increase the applicability of neural networks in fault recognition, especially for structures dominated by strike-slip faults. To briefly demonstrate the effectiveness of these changes, we use the neural network Net_syn_Wu from the open-source workflow by Wu et al. [68] for comparison. To compare fairly with the structural data augmentation, we use Equations (1)–(14) to generate 800 synthetic data–label pairs to train the neural network Net_syn_our (Figure 9b).

### 3.3.3. Network Training Based on Transfer Learning

In addition to manually interpreted data learning and synthetic data learning methods, transfer learning approaches have also become popular in recent years [74–78]. In this paper, we pretrain the network using the previously mentioned 800 synthetic seismic data (Equations (1)–(14)). Based on the pre-trained network, we fix most of the parameters in the network and fine-tune the last layer to further learn new data features as suggested by Wei et al. [118]. As shown in Figure 9c, the same 400 × 400 × 400 field data in the previously mentioned work area A (Figure 10a) are truncated into 100 pairs of 128 × 128 × 128 data–labels to fine-tune the network Net_transfer.

### 3.3.4. Network Training Based on Structural Data Augmentation

Finally, we use the structural data augmentation framework proposed in this paper to augment the same 400 × 400 × 400 field data from the work area A (Figure 10a). As shown in Figure 9d, 800 pairs of 128 × 128 × 128 data–labels generated by our framework are used to train the network Net_stru_aug.

## 4. Results

In this section, we demonstrate the applicability of our structural data augmentation method using field seismic data obtained from three different work areas in the Tarim Basin, China (work areas A, B, and C). Affected by the deep geological environment, the field data selected from the ultra-deep section (over 7000 m depth) contain complex structural features. In particular, multi-stage and multi-scale strike-slip faults are developed in these ultra-deep strata, which are difficult to characterize effectively by conventional methods [8]. We compare our fault prediction results with three recently popular machine

learning methods (manually interpreted data learning [62], synthetic data learning [68], and transfer learning [118]) and two classical conventional methods (coherence method [19] and curvature method [25]).

Figure 10 demonstrates the 3D full view and 2D slice details of the fault identification results in work area A. In work area A, our structural data augmentation method and Net_manual perform well, especially for flower-like strike-slip fault structures in detailed slices (Figure 10b,c). A few imperfections in the Net_manual may be due to the bias of the hand interpretation (e.g., the green rectangle in Figure 10c). The performance of Net_transfer (Figure 10d) is improved compared to that of Net_syn_our (Figure 10f). Transfer learning effectively reduces lineaments in the results that may not be related to faults (e.g., the green rectangular area in Figure 10d,f). However, the fault recognition results obtained by Net_transfer are still imperfect. Both Net_syn_Wu and Net_syn_our extract misinformation unrelated to faults (Figure 10e,f). This misinformation indicates that the synthetic data contain only limited ideal geological information and cannot fully bridge the gap between synthetic data and field seismic data. However, it is difficult for Net_syn_Wu to extract complete faults (e.g., the flower structure in the green rectangular area in Figure 10e). These incomplete fault characterization results may stem from the fact that some geological features have been simplified or omitted in the conventional structural modeling framework (especially for strike-slip faults). Coherence and curvature can produce relatively complex fault network images, but they are still disturbed by noise and makes it difficult to identify the fault structure completely (Figure 10g,h).

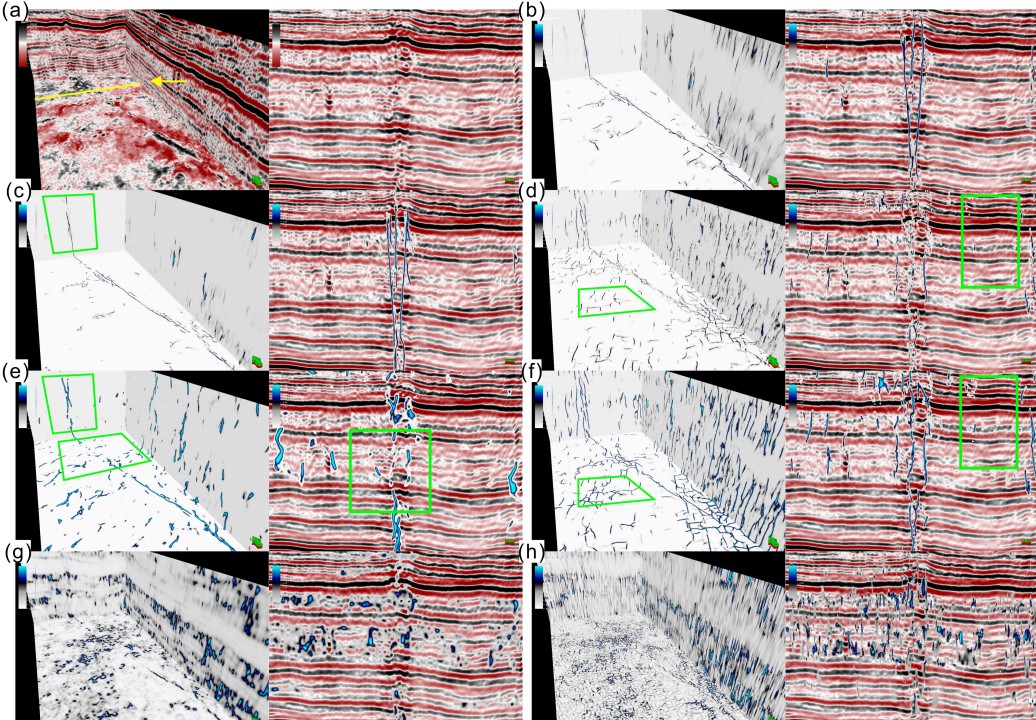

**Figure 10.** Fault recognition results of work area A. (**a**) Field seismic data. The yellow line represents the position of the detailed slice. (**b**) Detected faults by our structural data augmentation method. (**c**) Detected faults by Net_manual [62]. (**d**) Detected faults by Net_transfer [118]. (**e**) Detected faults by Net_syn_Wu [68]. (**f**) Detected faults by Net_syn_our. (**g**) Detected faults by Coherence [19]. (**h**) Detected faults by Curvature [25].

Figure 11 demonstrate the 3D full view and 2D slice details of the fault identification results in work area B. As shown in Figure 11b, our structural data augmentation method can effectively identify major faults. However, the results also contain some errors that may not be related to the fault (e.g., the green rectangular area in Figure 11b). These errors may be due to the differences between different work areas, indicating that our method

may still be imperfect in generalization ability. As shown in the green rectangular area in Figure 11c, Net_manual cannot extract complete faults. This incomplete result illustrates that a network trained with only the interpreted data from one work area may produce errors in cross-work-zone applications. As shown in Figure 11d,f, although the transfer learning method can reduce the error while preserving the main fault, the effect of fine-tuning is limited, and the result still has some misinformation (e.g., the green rectangular area in Figure 11d,f). As shown in Figure 11e, the results of Net_syn_Wu not only contain misinformation, but also the main fault is discontinuous (e.g., the green rectangular area in Figure 11e). This result shows that the geological information accommodated by the theoretical structural models is limited and can seriously affect the effectiveness of machine learning. The richer the geological information contained in the training data, the stronger the applicability of the trained network. Coherence and curvature are severely affected by noise when dealing with such ultra-deep complex seismic data (Figure 11g,h).

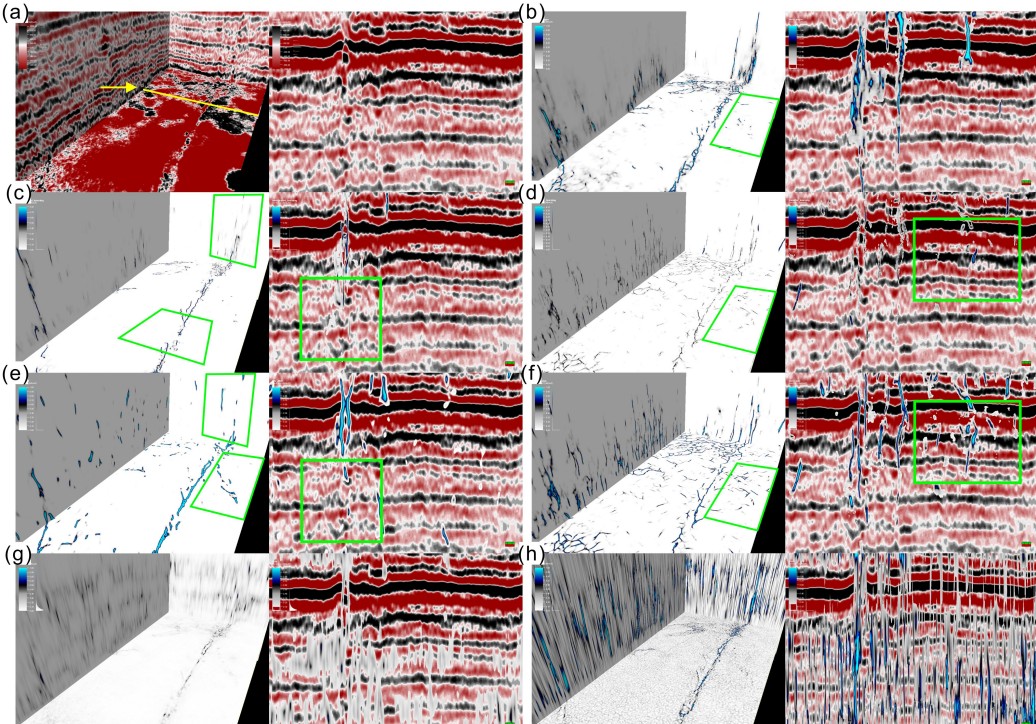

**Figure 11.** Fault recognition results of work area B. (**a**) Field seismic data. The yellow line represents the position of the detailed slice. (**b**) Detected faults by our structural data augmentation method. (**c**) Detected faults by Net_manual [62]. (**d**) Detected faults by Net_transfer [118]. (**e**) Detected faults by Net_syn_Wu [68]. (**f**) Detected faults by Net_syn_our. (**g**) Detected faults by Coherence [19]. (**h**) Detected faults by Curvature [25].

Figure 12 demonstrates the 3D full view and 2D slice details of the fault identification results in work area C. In work area C, both our structural data augmentation method and Net_manual extract some misinformation (Figure 12b,c). However, the extraction of secondary faults by Net_manual is not as good as our structural data augmentation method (e.g., the green rectangular regions in Figure 12c). These errors indicate that only using conventional augmentation methods from computer vision is insufficient to train the neural network effectively for fault extraction. As shown in Figure 12d,f, Net_transfer effectively reduces some misinformation in the results of Net_syn_our. However, transfer learning also undermines the integrity of the main fault extraction (e.g., the green rectangular area in Figure 12d). As shown in Figure 12e, Net_syn_Wu yields good fault identification results but simultaneously extracts partial stratigraphic information (e.g., the green rectangular area in Figure 12e). Coherence and curvature highlight a portion of the fault network in the identification results of the field data but are severely affected by noise.

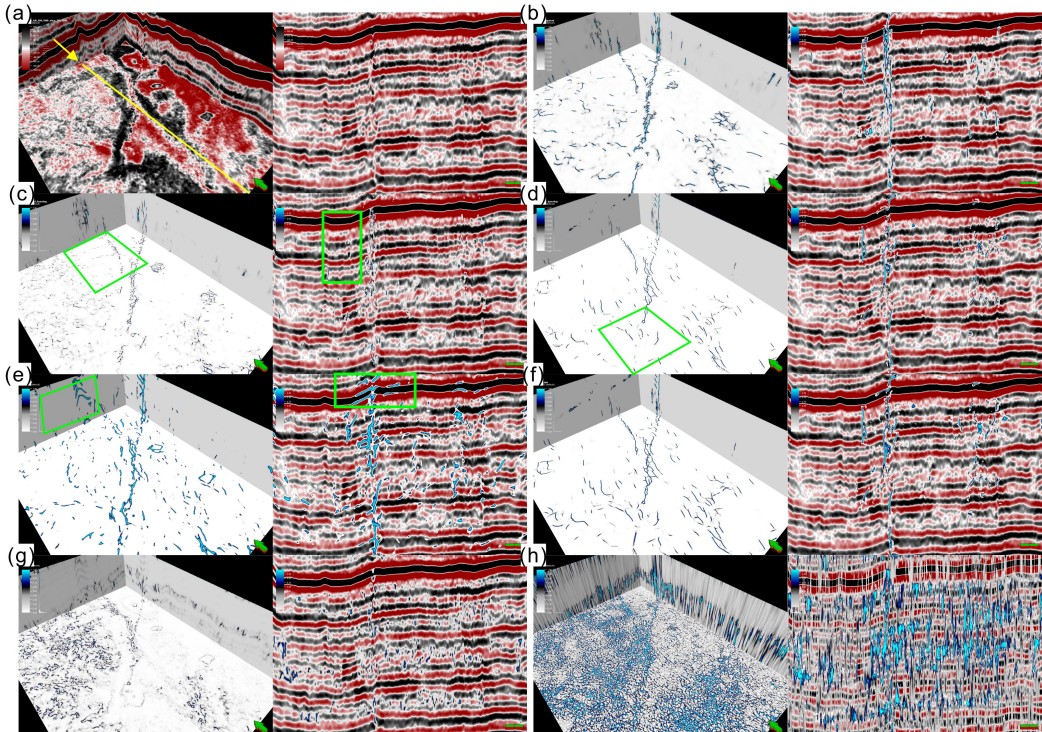

**Figure 12.** Fault recognition results of work area C. (**a**) Field seismic data. The yellow line represents the position of the detailed slice. (**b**) Detected faults by our structural data augmentation method. (**c**) Detected faults by Net_manual [62]. (**d**) Detected faults by Net_transfer [118]. (**e**) Detected faults by Net_syn_Wu [68]. (**f**) Detected faults by Net_syn_our. (**g**) Detected faults by Coherence [19]. (**h**) Detected faults by Curvature [25].

To further quantitatively evaluate the fault detection accuracy, we use results comparison tests [119] between the fault results extracted by different methods in work areas A, B, and C. Since the ground-truth fault behind these field seismic signals is unknown, and since each fault result contains false-positive and false-negative results due to different training sets and algorithms, it is a challenging task to evaluate the validity of different fault identification results accurately. Results comparison tests calculate relative performance metrics for other fault extraction results by assuming that one fault extraction result is ground truth. Taking the first row of the result comparison test as an example, the F1 scores of each method are calculated by setting our method as ground truth. Tables 1–3 show the F1 score of the results comparison tests in the work areas A, B, and C, respectively. From the results comparison tests, it can be observed that the results comparison coefficients between Net_syn_our and Net_transfer and between coherence and curvature are generally large. This correlation may indicate that the two groups of methods are algorithmically similar. Despite this unfavorable condition, our structural data augmentation method still achieves the highest scores in many cases, which means that most of the faults extracted by our method are also present in the results of other methods and are more likely to be real faults.

**Table 1.** Results Comparison Test with F1 Score in Work Area A.

| Fault Detection Methods | Ours | Net_manual [62] | Net_transfer [118] | Net_syn_our | Net_syn_Wu [68] | Coherence [19] | Curvature [25] |
|---|---|---|---|---|---|---|---|
| Ours | 1.0 | **0.2858** | **0.1804** | 0.2825 | **0.2870** | 0.2518 | 0.0931 |
| Net_manual [62] | 0.2858 | 1.0 | 0.1145 | 0.1315 | 0.2101 | 0.1054 | 0.0432 |
| Net_transfer [118] | 0.1804 | 0.1145 | 1.0 | **0.4418** | 0.1711 | 0.1287 | 0.0599 |
| Net_syn_our | 0.2825 | 0.1315 | **0.4418** | 1.0 | 0.2389 | **0.3133** | **0.1142** |
| Net_syn_Wu [68] | **0.2870** | 0.2101 | 0.1711 | 0.2389 | 1.0 | 0.2073 | 0.0781 |
| Coherence [19] | 0.2518 | 0.1054 | 0.1287 | **0.3133** | 0.2073 | 1.0 | **0.1782** |
| Curvature [25] | 0.0931 | 0.0432 | 0.0599 | 0.1142 | 0.0781 | **0.1782** | 1.0 |

**Table 2.** Results Comparison Test with F1 Score in Work Area B.

| Fault Detection Methods | Ours | Net_manual [62] | Net_transfer [118] | Net_syn_our | Net_syn_Wu [68] | Coherence [19] | Curvature [25] |
|---|---|---|---|---|---|---|---|
| Ours | 1.0 | **0.2008** | **0.1772** | **0.3372** | **0.2125** | **0.3221** | **0.2030** |
| Net_manual [62] | 0.2008 | 1.0 | 0.1175 | 0.1398 | 0.1842 | 0.1267 | 0.0571 |
| Net_transfer [118] | 0.1772 | 0.1175 | 1.0 | **0.4335** | 0.1381 | 0.1299 | 0.0703 |
| Net_syn_our | **0.3372** | 0.1398 | **0.4335** | 1.0 | 0.1859 | 0.2984 | 0.1956 |
| Net_syn_Wu [68] | 0.2125 | 0.1842 | 0.1381 | 0.1859 | 1.0 | 0.1576 | 0.0929 |
| Coherence [19] | 0.3221 | 0.1267 | 0.1299 | 0.2984 | 0.1576 | 1.0 | **0.4259** |
| Curvature [25] | 0.2030 | 0.0571 | 0.0703 | 0.1956 | 0.0929 | **0.4259** | 1.0 |

**Table 3.** Results Comparison Test with F1 Score in Work Area C

| Fault Detection Methods | Ours | Net_manual [62] | Net_transfer [118] | Net_syn_our | Net_syn_Wu [68] | Coherence [19] | Curvature [25] |
|---|---|---|---|---|---|---|---|
| Ours | 1.0 | **0.1968** | **0.1209** | **0.2215** | 0.1882 | 0.1070 | 0.0509 |
| Net_manual [62] | 0.1968 | 1.0 | 0.0807 | 0.1505 | 0.1540 | 0.1174 | 0.0523 |
| Net_transfer [118] | 0.1209 | 0.0807 | 1.0 | **0.4346** | 0.0979 | 0.0411 | 0.0149 |
| Net_syn_our | **0.2215** | 0.1505 | **0.4346** | 1.0 | 0.1942 | 0.0949 | 0.0371 |
| Net_syn_Wu [68] | 0.1882 | 0.1540 | 0.0979 | 0.1942 | 1.0 | **0.2078** | **0.0982** |
| Coherence [19] | 0.1070 | 0.1174 | 0.0411 | 0.0949 | **0.2078** | 1.0 | **0.2637** |
| Curvature [25] | 0.0509 | 0.0523 | 0.0149 | 0.0371 | 0.0982 | **0.2637** | 1.0 |

## 5. Conclusions

In this study, we describe a method called structural data augmentation to obtain diverse field-like seismic data. Many unique virtual structures can be generated in the seismic data by randomly combining multiple parameters. The purpose of our work is to bridge the gap between training data and field data, enhance the generalization ability of seismic data, and obtain corresponding accurate labels to train neural networks. Compared with manual interpretation data learning, synthetic data learning, and transfer learning methods, our approach (1) combines information from theoretical models and knowledge of manual interpretation to provide training data with high generalization capability, (2) bridges the gap between training data and field seismic data, and (3) achieves better results with only a simple U-net. These comparisons illustrate that our method is learning-friendly and may have the prospect of broad industrial application. The fault identification results from field data prove that the proposed method can be effectively applied to seismic data interpretation, and learning sets similar to field data and rich in geological information can improve the applicability of machine learning.

Although we emphasize the flexibility and generalization ability of our workflow in seismic data augmentation throughout the paper, this method still has limitations. The structural data augmentation method does not increase the diversity of seismic waveforms and noise types, which may lead to overfitting. Incorporating more types of seismic noise and utilizing different waveform decomposition methods in future work may further

increase the generalization ability of the method. The virtual faults added to the seismic data can produce sharp discontinuities. In practical applications, moderate smoothing may help to simulate more realistic faults in seismic images because smoothing can blur sharp discontinuities near faults. The fault recognition results in this study are only achieved by a simple U-net, and the use of more complex neural network architectures in future work may further improve the fault recognition results. In this study, there is currently no better way to quantitatively analyze the accuracy of identification results from field data. Although synthetic data have ground truth fault labels, it is not fair to utilize synthetic data for accuracy analysis. In future work, it may be a feasible option to quantitatively evaluate fault identification results from field data using well logging data.

**Author Contributions:** Conceptualization, S.W.; methodology, S.W.; coding, S.W. and X.S; validation, X.S; writing—original draft preparation, S.W.; writing—review and editing, S.W., X.S., Z.C. and Y.C.; visualization, S.W., X.S. and Y.C.; project administration, Z.C.; funding acquisition, Z.C. All authors have read and agreed to the published version of the manuscript.

**Funding:** This work was supported by the Strategic Priority Research Program of the Chinese Academy of Sciences, China (Grant No. XDA14010302).

**Institutional Review Board Statement:** Not applicable.

**Informed Consent Statement:** Not applicable.

**Data Availability Statement:** The training sets can be found at https://github.com/sixu0/Fault_aug, accessed on 1 September 2020.

**Acknowledgments:** We thank the Editors and reviewers for their helpful comments and suggestions. We gratefully acknowledge the Northwest Oilfield Branch Company, Sinopec, for generously supplying the field data and manually interpreted datasets for this study.

**Conflicts of Interest:** The authors declare no conflict of interest.

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
