# Peer review of "Structural Augmentation in Seismic Data for Fault Prediction"

_applsci, doi:10.3390/app12199796_

Round 1
Reviewer 1 Report
This paper proposes a novel data augmentation workflow to improve the network's performance on fault detection. The manuscript is well written. My comments are as follows:
1. The title mentions that the augmentation workflow is for ultra-deep fault prediction. But my understanding is that the method is very general to all kinds of faults and should perform well. Looks like it is only because the test data has ultra-deep faults. If it is, maybe you could remove "for ultra-deep faults" from the title, since it is not only designed for ultra-deep faults.
2. The term cross-validation isn't appropriate here. Even though the cited paper used that word, I don't think this kind of comparison is called cross-validation based on its definition. Please remove it to avoid confusion. You could use some other terms or even create one for this kind of comparison. A simple "results comparison" should also work in this case.
3. Could the authors explain a little bit on the difference between their workflow and the workflow from Wu et al. 2019? If it is possible, please summarize it in several sentences to highlight the improvement so that readers could better understand.
Reviewer 2 Report
The paper addresses the problems related to the quantity & quality of datasets for deep learning-based automatic interpretation techniques for fault prediction. It introduces a new data augmentation technique, named structural data augmentation, based on geological structure theories, that can help to add virtual folds & faults in existing seismic data resulting in improving the diversity & generalization ability of datasets.
The introduced approach for seismic data augmentation is much better than general approaches for image data augmentation since the approach generates more "natural" synthetic data using sound theoretical techniques. The paper is well written and well organized, and the theoretical and experimental results demonstrated are correct and discussed properly.
Some suggestions for improvement:
Lines 5-6: the sentence 'In this paper...' is not clearly connected to the previous sentences in the paragraph.
Lines 54-55: the sentence 'Therefore, ...' is not related to the paragraph.
The sentence ' ... we propose ...' in Lines 83-84 is repeated in line 95.
A better title for Section 5 is just "Conclusions" since the results are discussed in the previous section.
Reviewer 3 Report
The manuscript presents a relevant data augmentation technique to improve the modeling of structural systems. The approach is current and essential, especially for reservoir modeling. The textual argument is well structured. However, some methodological issues need to be better detailed so that the manuscript becomes more adequate.
Some specific points that need to be clarified:
1- the authors mention the use of seismic data. This information is generic. It is unclear which seismic attributes were used;
2- in the topic of methods, it is not clear the amount of information originally available for each label used. Likewise, the amount of data augmentation for each label is unclear;
3- Would the proposed method only be valid for ultra-deep faults? Why would the method not be suitable for shallow or intermediate structures, for example?
4- the validation of the results is extremely deficient since it is based on the comparison between methods. The most adequate evaluation of the results between different methods could be done using a synthetic model with a well-defined solution;
5- I suggest that the authors observe a little more about the central object of the manuscript: if the focus would be on increasing labeled data or if the focus would be on the modeling process itself. The evaluation of data augmentation should follow a different route, including observing the efficiency of labels built by data augmentation. An explanation of this should be part of the topic related to the evaluation of results.